# An RDL Modeling and Thermo-Mechanical Simulation Method of 2.5D/3D Advanced Package Considering the Layout Impact Based on Machine Learning

**DOI:** 10.3390/mi14081531

**Published:** 2023-07-30

**Authors:** Xiaodong Wu, Zhizhen Wang, Shenglin Ma, Xianglong Chu, Chunlei Li, Wei Wang, Yufeng Jin, Daowei Wu

**Affiliations:** 1Department of Mechanical & Electrical Engineering, Xiamen University, Xiamen 361005, China; wuxiaod1987@stu.xmu.edu.cn (X.W.); zhizhenwang@xmu.edu.cn (Z.W.);; 2School of Integrated Circuits, Peking University, Beijing 100871, China; w.wang@pku.edu.cn; 3School of Electronic and Computer Engineering, Peking University Shenzhen Graduate School, Shenzhen 518055, China; yfjin@pku.edu.cn; 4Xi’an Microelectronic Technology, Xi’an 710071, China; wudaowei1220@163.com

**Keywords:** redistribution layer, layout impact, machine learning, thermo-mechanical simulation, equivalent material properties

## Abstract

The decreasing-width, increasing-aspect-ratio RDL presents significant challenges to the design for reliability (DFR) of an advanced package. Therefore, this paper proposes an ML-based RDL modeling and simulation method. In the method, RDL was divided into blocks and subdivided into pixels of metal percentage, and the RDL was digitalized as tensors. Then, an ANN-based surrogate model was built and trained using a subset of tensors to predict the equivalent material properties of each block. Lastly, all blocks were transformed into elements for simulations. For validation, line bending simulations were conducted on an RDL, with the reaction force as an accuracy indicator. The results show that neglecting layout impact caused critical errors as the substrate thinned. According to the method, the reaction force error was 2.81% and the layout impact could be accurately considered with 200 × 200 elements. For application, the TCT maximum temperature state simulation was conducted on a CPU chip. The simulation indicated that for an advanced package, the maximum stress was more likely to occur in RDL rather than in bumps; both RDL and bumps were critically impacted by layouts, and RDL stress was also impacted by vias/bumps. The proposed method precisely concerned layout impacts with few resources, presenting an opportunity for efficient improvement.

## 1. Introduction

Redistribution layer (RDL) is one of the most important components of advanced packages like Chip on Wafer on Substrate (CoWoS) [1], Foveros [2], X-Cube [3] and System on Integrated Chips (SoIC) [4,5], especially the multilayer RDL interposer-based CoWoS-R [6], FO-PLP [7] and chiplet-based architectures [8,9]. To enable high-bandwidth signal processing between chiplets [10], several techniques, such as dual damascene processing, semi-additive processing (SAP) and polymer damascene processing, have been introduced. These techniques narrow the RDL trace width/pitch to as small as two microns [11,12] and even sub-microns [13]. To enhance the electricity transmission capacity, traces become thicker, with an aspect ratio of up to 4.2 [14], which is very different from traditional shell-like thin traces. Furthermore, wafer-level integrations can stack up to five metal layers [1], with an area up to 1200 mm^2^ [15] or even as extensive as 2500 mm^2^ [1]. As a result, RDLs in advanced packages become extremely complex in shape and behavior, posing great challenges in accurately capturing their thermo-mechanical characteristics. Additionally, in advanced packages, substrate is extremely thinned to several microns or even sub-microns [16] which are similar to or even thinner than RDL, making the RDLs more prominent compared to other package components [17,18]. For this reason, RDLs are playing crucial roles in package thermo-mechanical characteristics.

Integrated packages face severe thermo-mechanical risks due to the coefficient of thermal expansion (CTE) mismatch between materials; the accelerated thermal cycling test (TCT) [19] is an important approach for assessing their reliability. As a low-cost alternative to testing, finite element analysis (FEA) simulation under TCT conditions has become a significant research area for design for reliability (DFR). Balancing between efficiency and accuracy of simulation is a critical issue in this field.

For traditional integrations, due to the thicker substrate and larger trace width/pitch, axisymmetric FEA models can meet the efficiency and accuracy requirements for DFR. Using a 2D axisymmetric model, Lee et al. [20], Che et al. [21] and Machani et al. [22] simplified the RDLs into homogeneous rectangles and refined the outermost bump, then evaluated the fatigue life of the bump through transient simulation. Using a 3D axisymmetric model, Lee et al. [23] and Che et al. [24] simplified the RDL into homogeneous films and refined the corner solder, thus assessing the reliability of stacked-chip packages. However, advanced packaging has different structural characteristics.

For advanced packages, RDL plays a more significant role in thermo-mechanical reliability. Therefore, more precise RDL modeling approaches are required for DFR of advanced packages. Dividing the RDL and then employing material equivalization is a widely used strategy to improve the accuracy. In this strategy, division size and equivalization method are crucial for efficiency and accuracy. Where efficiency is concerned, RDL is divided into large-size regions, and there are three typical material equivalization methods. The first method, employed by Wang et al. [25], involves calculating the isotropic materials based on the copper ratio within each region. The second method, used by McCaslin et al. [26], is determining the anisotropic equivalent material properties of regions consisting of simple shape traces through composite material mechanics. The third one, utilized by Valdevit et al. [27], is the morphology-based approach, where regions are categorized into “lines”, “vias” or “web” types, and then specific mathematical models are developed to compute the material properties for each type of region. Where accuracy is concerned, RDL is divided into small-size blocks, and there are also three typical material equivalization methods. The first method, employed by Lien et al. [28], involves calculating the isotropic equivalent material properties based on the metal percentage within each block. The second method, used by Lee et al. [29], focuses on calculating the anisotropic material properties and the equivalent reference temperature of blocks using composite material mechanics. The third one, introduced by Gibson [30] and Lee et al. [29], is a simulation-based approach that calculates the anisotropic equivalent material properties of blocks containing complex traces. To achieve higher accuracy, Yaddanapudi [31] divided the RDL into extremely small-size pixels and directly defined each pixel as either metal or dielectric; a high-fidelity structural response of RDL was achieved by consuming a large amount of solving resources. In general, existing work indicates that there is a trade-off between division sizes and the material equivalization method. The key to balancing efficiency and accuracy lies in obtaining the anisotropic equivalent material properties quickly and accurately under a larger division size.

Recently, machine learning (ML) has shown the capability of extracting interpretable models from scientific data automatically [32]; it has been increasingly employed in the design and control of robots [33], actuators [34] and pumps [35]. Material equivalization of composite materials in 3D and 2D structures is a major aspect of ML research. Regarding 3D, Dai et al. [36] used feature matrixes to digitally represent the lattice orientations of 3D structures, and then created a graph-Artificial Neural Network (ANN) to predict the structural performance. Regarding 2D, Liu et al. [37], Ye et al. [38] and Gong et al. [39] used a high-resolution grayscale matrix to digitalize the planar structures, then constructed convolutional neural networks (CNNs) to quickly predict the equivalent mechanical properties. Due to the accurate and fast predictive capabilities of models created based on ML, it has been introduced into the modeling and simulation of integrated packages. For example, Selvanayagam et al. [40,41] partitioned an 8-RDLs interposer into 3 × 3 regions, then established surrogate models that linked the copper ratio in each layer within each region to the warpage in TCT condition, ultimately optimizing the global package warpage.

In this work, we applied ML to material equivalization and developed an RDL modeling and simulation method for DFR of advanced packages. In this method, an RDL was divided into equal-size blocks. To describe the layout within each block, we further subdivided each block into pixels and represented each pixel using the metal percentage. Consequently, each block could be represented as a tensor composed of pixel metal percentages, and the entire RDL was digitalized as a collection of such block tensors. An ANN was constructed using either fully connected neural networks (FCNNs) or convolutional neural networks (CNNs) to model the relationship between the block tensor and the corresponding equivalent material properties. Subsequently, the ANN was trained using a small subset of tensors with the equivalent material properties obtained through finite element analysis (FEA) as labels, and then the material surrogate model was constructed. Lastly, all blocks were transformed into FEA elements and assigned the material properties predicted by the model, completing the modeling and simulations of the RDL.

To validate the efficiency and accuracy of the proposed method, line bending simulations were conducted for a 21.6 × 21.6 mm^2^ RDL. The detailed fine mesh model served as the benchmark, with the reaction force being regarded as the accuracy indicator. The results show that as the substrate thickness reduced, the reaction force error of the traditional method that neglected layout impact increased sharply, indicating that critical layout impact must be considered in the simulation of the advanced package. When the proposed method was used with a substrate thickness of 50 μm, the reaction force error was 2.81%. The layout impact could be accurately estimated utilizing only 200 × 200 elements; the number of elements was only 1/15 of the traditional approach.

As an application case, a simulation was introduced to investigate the thermo-mechanical response of a 2.5D integrated CPU chip at the maximum temperature state in TCT. The results indicate that for advanced packages, the maximum stress was more likely to occur in the RDL, which is different from the traditional integrations, where the maximum stress occurs in the bumps. It also reveals that the stress of both the RDL and bumps is significantly impacted by the layout. Moreover, it is observed that the stress in RDL is particularly impacted by vias and bumps in adjacent layers.

The method precisely concerns these layout-related impacts with minimal resources and time, presenting an opportunity to improve the efficiency of advanced package DFR.

## 2. Methods

The general workflow of the proposed modeling method is shown in Figure 1 and consists of the following steps:

Step 1: Two-level RDL digitalization. This step consists of global and local discretization. In the global level, the package-level RDL pattern is divided into equal-size RDL blocks containing part of the traces, and there are *Q_x_* blocks along the X direction and *Q_y_* blocks along the Y direction. To capture the layout feature within blocks, in the local level, each RDL block is divided into pixels with a uniform size. The subdivision results in *PQ_x_* pixels along the X direction and *PQ_y_* pixels along Y; the pixel value is defined as the metal percentage within the pixel region. Consequently, the RDL block can be represented by a *PQ_x_* × *PQ_y_* tensor, and the package-level RDL can be digitally represented by *Q_x_* × *Q_y_* tensors.

Step 2: ANN Training dataset preparing. This step prepares the ANN training dataset by randomly selecting a small subset of RDL blocks. The dataset consists of input tensors and labels representing the anisotropic equivalent material properties of each block obtained through 3D FEA simulations [16], including Young’s modulus, Poisson’s ratios, shear modulus, thermal conductivities and CTEs in all directions.

Step 3: ANN-based surrogate model establishing. This step establishes an ANN-based material surrogate model. The input data of the surrogate model include the RDL block tensor and the output data include all equivalent properties. The surrogate model employs ANN and is trained with the dataset prepared in Step 2. Once established, the ANN eliminates the need for repeating Steps 2 and 3; it would be highly advantageous for the repeated design iteration process.

Step 4: Equivalent properties predicting. All equivalent properties for all the RDL blocks generated in Step 1 are predicted with the surrogate model established in Steps 2 and 3.

Step 5: Global FEA model building. All RDL blocks are transformed into 3D hexahedral solid elements to construct the global RDL FEA model and assigned the material properties predicted in Step 4. Each element is built by sweeping the geometry face of the RDL block, with the sweep length equal to the RDL thickness; therefore, all elements are the same size. The model consists of *Q_x_* × *Q_y_* elements encompassing the entire RDL and can also serve as a part of the package-level FEA model.

As the method is implemented, a series of programs are established, including the ANN implemented with Pytorch, the main framework developed in Python and the FEA simulator powered by ANSYS 18.2. The RDL block creating processes are implemented with C++. In addition, the C# developed program is used as the graphical post-processor.

### 2.1. ANN Architectures

Activation function. The rectified linear unit (ReLU) is selected as the activation function, shown by (1), (2):(1)f (x)=max(0, x)
(2)$$f\prime \left(x\right)=\{\begin{array}{c} 1,\;x\;\geq \;0\\ 0,\;x\;<\;0 \end{array}$$

Loss Function. Cross-entropy loss is selected as the loss function, shown by (3):(3)$$\mathrm{Loss}=-\overset{n}{\underset{i=1}{\sum }}w_{i}y_{i}\log\hat{y_{i}}$$
where *n* is the number of samples, *y_i_* represents the true value of the *i*-th sample, $\hat{y_{i}}$ represents the predicted value of the *i*-th sample and *w_i_* represents the weight of the *i*-th sample. The loss for a batch prediction is shown by (4):(4)$$\mathrm{Loss}=-\frac{1}{batch}\;\sum_{j=1}^{batch}\sum_{i=1}^{n}w_{ij}y_{ij}\log\hat{y_{ij}}$$
where *batch* is the number of predicting procedures for one predicted batch. For the *j*-th predicting procedure of the batch, *y_ij_* represents the true value of the *i*-th sample, $\hat{y_{ij}}$ represents the predicted value of the *i*-th sample and *w_i_* represents the weight of the *i*-th sample.

Optimizer. The stochastic gradient descent (SGD) was adopted to optimize the neural network, and momentum was considered during the optimization process. Learning rate and momentum parameters were adjusted to fine-tune the optimization process, as indicated in Equations (5) and (6):(5)$$v_{t}{=m\;\times \;v}_{t\;-\;1}\;-\;lr\;\times \;g$$
(6)$$w_{t}{=w}_{t\;-\;1}{+v}_{t}$$
where *t* is the current iteration step number, *t* − 1 is the number of the previous step, *lr* represents the learning rate, *g* is the gradient and m means the momentum. *v_t_* and *w_t_* represent the velocity and weight in the current iteration step, respectively. *v_t_*_−1_ and *w_t_*_−1_ represent the velocity and weight in the previous step, respectively.

ANN networks. The full-connected neural network (FCNN) and convolution neural network (CNN) are created. As shown in Figure 2, the FCNN includes an input layer, output layer and several hidden layers. The parameters of the input layer are represented by the vector shown by (7) and the size of the input vector is shown by (8):(7)$${I=[I}_{1}^{\left[0\right]}{,\;I}_{2}^{\left[0\right]}{,\ldots ,I}_{p}^{\left[0\right]}{,\ldots ,I}_{Q_{in}}^{\left[0\right]}]$$
(8)$$Q_{in}{=PQ}_{x}{\;\times \;PQ}_{y}$$
where $I_{p}^{\left[0\right]}$ is the metal percentage of the *p*-th pixel, *Q_in_* is the vector size which is equal to the number of pixels in a single RDL block and *PQ_x_* and *PQ_y_* represent the number of RDL block pixels along the X and Y directions, respectively. The output vector of the FCNN includes all 15 anisotropic equivalent material properties of the RDL block, shown by (9):(9)$${O=[E}_{x}{,\;E}_{y}{,\;E}_{z}{,\;\mu }_{xy}{,\;\mu }_{xz}{,\;\mu }_{yz}{,\;G}_{xy}{,\;G}_{xz}{,\;G}_{yz}{,\;k}_{x}{,\;k}_{y}{,\;k}_{z}{,\;A}_{x}{,\;A}_{y}{,\;A}_{z}]$$

The hidden layer can be represented by vector as (10):(10)$$A^{\left[n\right]}{=[A}_{1}^{\left[n\right]}{,\;A}_{2}^{\left[n\right]}{,\ldots ,A}_{i}^{\left[n\right]}{,\ldots ,A}_{H_{n}}^{\left[n\right]}]$$
where $A_{i}^{\left[n\right]}$ is the value of the *i*-th node and *Q_n_* is the number of nodes of the *n*-th hidden layer.

Shown in Figure 3, the CNN consists of the convolution network and the full-connected network; the input parameter is the tensor with a size of [*PQ_x_*, *PQ_y_*, 1], and the output vector is the same as (7). The convolution network contains convolution layers and max pooling layers, and the tensor is transformed through layers. The tensor size is [*C_nx_*, *C_ny_*, *C_nz_*] after the *n*-th convolution layer and [*P_nx_*, *P_ny_*, *P_nz_*] after the *n*-th max pooling layer. The output tensor of the convolution network can be compressed into a vector of size *Q_Lin_*.
(11)$$Q_{Lin}{=P}_{nx}{\;\times \;P}_{ny}{\;\times \;P}_{nz}$$
where *P_nx_*, *P_ny_* and *P_nz_* are the output tensor size in the X, Y and Z directions, respectively.

### 2.2. Training Dataset Augmentation

To augment the training dataset of blocks, the geometric symmetry and data transformations were employed. The initial block with anisotropic properties (E*_x_* = E*_x0_*, E*_y_* = E*_y0_*) is shown in Figure 4a, and it can be rotated by 90°, 180° and 270°, as indicated in Figure 4b–d, respectively. When the angle of rotation is 90° or 270°, the anisotropic properties are transformed to E*_x_* = E*_y0_*, E*_y_* = E*_x0_*. The initial block and its rotated versions can also be flipped in the Y direction, as shown in Figure 4e–h, while keeping the same anisotropic properties. Eight unique blocks can be generated from the initial one without requiring additional FEA solutions.

## 3. Result and Discussions

### 3.1. Method Validation

The two-layer model containing RDL and a silicon substrate is shown and established for validation in Figure 5. The model consists of the substrate layer and the RDL (the M1 layer); the materials involved in the simulations are listed in Table 1, and all materials are assumed to be linear elastic. The substrate is made up of Si; its thickness (T_B_) is set as a variable parameter, with a range of values from 20 μm to 500 μm. The RDL consists of Cu and PI; the thickness (T_R_) is 10 μm. The layout pattern of the M1 layer is shown in Figure 6 and the domain direction of traces is the X direction.

The validation of the adaptability to the mechanical loads is based on the parallel trace bending load case (CaseP) and vertical trace bending load case (CaseV) shown in Figure 5. In the CaseP, nodes on the X = 0 face were constrained in the X, Y and Z directions, and nodes on the top line of the X = Length face were set at Z = −0.5 mm. In the CaseV, the constraints were set on the Y = 0 face, and the displacement load was set on the top line of the Y = Width face.

As the benchmark, a detailed RDL fine mesh model consisting of 640,365 nodes and 606,482 elements was established with a substrate thickness of 50 μm. The solving process cost 176 s. The stress distributions are depicted in Figure 7; stress concentrations can be observed at discontinued positions, indicating the critical impact from the layout.

The sums of the Z-direction reaction forces of all nodes applied and the displacement loads in CaseP and CaseV are *FR_DP_*: −7.174 × 10^−3^ N and *FR_DV_*: −6.464 × 10^−3^ N, respectively. The reaction force errors serve as accuracy indicators, as shown in (12).
(12)$$\{\begin{array}{c} {Error}_{P}=\left|{(FR}_{P}{\;-\;FR}_{DP})\;/{\;FR}_{DP}\right|\;\times \;100\%\\ {Error}_{V}=\left|{(FR}_{V}{\;-\;FR}_{DV})\;/{\;FR}_{DV}\right|\;\times \;100\% \end{array}$$
where *Error_P_* and *Error_V_* are reaction force errors for CaseP and CaseV, respectively. *FR_P_* and *FR_V_* are reaction forces to be evaluated for CaseP and CaseV, respectively.

For the proposed method, considering the efficiency and accuracy, we set the global mesh divisions at *Q_x_* = *Q_y_* = *Q* = 200 and the local pixel division at *PQ_x_* = *PQ_y_* = *PQ* = 15, as well as building 40,000 RDL blocks. *Q_t_* blocks were randomly selected as the ANN training dataset, the default value of *Q_t_* was 1000, and the testing dataset contained all 40,000 blocks. An FCNN with two hidden layers was employed for the ML-based method. As depicted in Figure 2, the first hidden layer contained 400 nodes (H1 = 400), while the second one had 300 nodes (H2 = 300). The learning rate was set at *lr* = 0.02, the momentum was set at *m* = 0.5 and the training epoch was set at *q* = 250. The training and testing processes were run using a GPU of NVidia RTX 3070Ti; the average predicting time of all blocks was 3.652 s. The calculated material properties are shown in Figure 8 and the loss variation is shown in Figure 9.

The calculated material properties indicate the material properties predicted by the ML-based method were very close to those obtained with the simulation-based method, which demonstrates the accuracy of the ANN-based surrogate model. Additionally, there was a notable difference between Young’s modulus along the trace direction (E*_x_*) and along the perpendicular direction (E*_y_*). Moreover, the reduction in E*_y_* is obvious within the gap area between traces, indicating the critical impact of RDL layout on the mechanical behavior of RDL.

The stress distribution shown in Figure 10 is consistent with the result of the detailed fine mesh model presented in Figure 7. Additionally, the reaction force errors for CaseP and CaseV are 1.72% and 2.81%, respectively, validating the precision of this method. Furthermore, the proposed approach only requires 200 × 200 FEA elements, whereas the detailed fine mesh model consists of 15 times more elements. This demonstrates that the approach contributes significant resource savings in simulation.

Furthermore, to verify the accuracy of thermo-mechanical simulations, a parallel trace warpage case (CasePT) and vertical trace warpage case (CaseVT) were created, as shown in Figure 11. Except for the 150 °C temperature, nodes on the X = 0 and Y = 0 faces were constrained in the X, Y and Z directions in the CasePT and CaseVT cases, respectively. Figure 12 and Figure 13 show the Z direction displacement and von Mises stress distributions of the cases; they present almost identical deformation and stress distributions as the detailed fine mesh model.

### 3.2. Key Factors Influence

The influence of the substrate thickness, global mesh division, training dataset size, and ANN architecture on the prediction process was investigated.

The influence of the substrate thickness.

Three simulation methods were performed, including the traditional layout-neglecting block-based method, the simulation-based method and the proposed ML-based methods, on seven substrate thickness values: 20, 50, 100, 200, 300, 400 and 500 μm. Figure 14 summarizes the reaction force errors of each case, demonstrating the fact that as the substrate thickness decreased, the impact of RDL layout became more and more critical; neglecting the layout features resulted in substantial errors.

The influence of the global mesh division.

Two more global division values of 100 and 300 were conducted to investigate the influence of the global division; the reaction force errors of different cases are summarized in Figure 15. The reaction force error of CaseP was 2.255% when *Q* = 100; it fell to 1.149% when *Q* = 200 and only 0.076% when *Q* = 300, illustrating that larger global mesh divisions can improve solution accuracy. When *Q* increased from 100 to 300, the solution time increased by a factor of 11.5 and the memory occupation increased by a factor of 6.59. This suggests that a higher global mesh division can achieve lower error, but it also increases the solution time and resource consumption.

Figure 16 depicts stress distributions at *Q* = 100 and *Q* = 300; the stress distribution shown in Figure 16a is similar to those shown in Figure 7 and Figure 10, indicating that the proposed method can accurately capture the layout impact shown in Figure 6 using only 100 × 100 elements. Moreover, larger *Q* values can capture more accurate pattern features, but more solving time and resources are needed. Regarding the block-based method, as shown in Figure 17, an obvious stress inhomogeneity was only captured until *Q* ≥ 600. Moreover, even when *Q* = 800, Figure 18 demonstrates that the accuracy could not match that of the proposed method. However, simulation costs became exorbitant.

The influence of the training dataset size.

ANNs were established using ten different dataset sizes ranging from 500 to 5000. Figure 19 summarizes the loss and training time. The prediction loss curves indicate that the prediction loss decreased as the size of the training dataset increased, but the effect of dataset size diminished. When the dataset size exceeded 2000, curves of prediction losses were nearly parallel to the *X*-axis, suggesting that even if the dataset size further increased, the prediction accuracy would not be improved. The prediction loss curves using the dataset augmentation were lower than the curves without dataset augmentation, indicating that the proposed dataset augmentation algorithm can improve the prediction accuracy.

The equivalent material properties predicted with the dataset augmentation algorithm are shown in Figure 20. The results indicate a heightened similarity to the simulation-based method results depicted in Figure 9.

The influence of the FCNN architectures.

Three different types of FCNNs with two hidden layers and varying numbers of hidden nodes were established. The training and prediction losses are summarized in Figure 21a. The prediction loss curves with more hidden layer nodes were lower, indicating that increasing hidden layer nodes resulted in improved prediction accuracy. However, the loss reduction in H1 from 800 to 1200 was smaller than that from 400 to 800, indicating the effect diminished as nodes increased. Additionally, Figure 21b illustrates the loss curves of the FCNNs with three hidden layers, showing that additional hidden layers could improve accuracy, but it required more training time.

Application of the CNN architectures.

Figure 22 summarizes the losses and training time for “1 CNN + 1 FCNN”, “1 CNN + 2 FCNNs” and “2 CNNs + 2 FCNNs”. The results demonstrate that while CNNs had a shorter training time, they exhibited significantly larger losses across all three architectures. The intermediate tensors during convolution and max pooling operations are shown in Figure 23, which indicates the fact that the key features of the initial block tensor were eliminated after those operations. The undesirable defeaturing has a negative impact on the prediction accuracy.

### 3.3. Large Area 2.5D-Integrated CPU Chip Thermo-Mechanical Simulation

As an application, a 2.5D-integrated CPU chip was constructed, as shown in Figure 24. All three RDLs were modeled using the proposed method; the division parameters were *Q* = 200, *PQ* = 15. The training dataset size of *Q_t_* = 1000 was utilized with data augmentation. Predictions were made using FCNNs with two hidden layers consisting of 400 and 300 nodes, respectively. In the package-level FEA model, the maximum element size was 0.12 mm; it contained 1,072,658 nodes and 625,648 elements.

The equivalent Young’s module distributions of the M1B and M2 layers are shown in Figure 25 and Figure 26, respectively. A good consistency with the simulation-based method is shown. The equivalent material property prediction times for the M1B and M2 layers are only 2.8211 s and 2.833 s, respectively.

To simulate the maximum temperature state of the TCT, a temperature load of 150 °C was applied and the bottom surface of the BGA was constrained in the X, Y and Z directions, as shown in Figure 27. The results obtained under this condition are shown in Figure 28, indicating the following information:

Firstly, it is found from Figure 28a that the maximum stress occurs in the M1 layer rather than bumps; this indicates that for 2.5D advanced packages, as the substrate thickness decreases and the trace width/pitch reduces, the thermo-mechanical risk of RDL becomes more significant. The conclusion for traditional integrations—that the maximum risk always occurs at bumps—is no longer accurate.

Secondly, Figure 28b,c demonstrate the fact that the stress distribution of RDL is critically impacted by layout; neglecting the layout impact would bring in critical errors, and the proposed method can accurately capture the impact.

Furthermore, it can also be found from Figure 28a that the location and magnitude of the maximum stress of bumps and TSVs are influenced by the layouts and surrounding structures, and it is different from the conventional understanding that the outermost bumps are the most vulnerable. Therefore, the impact of the vias and bumps layouts should also be considered in the simulation.

Finally, the detailed view of the maximum stress region of M1 layer in Figure 28d illustrates that the stress on RDL is not only influenced by the RDL layout but also critically impacted by the vias and bumps within adjacent layers.

To investigate the reasons behind the local influence of RDL geometric features, adjacent vias and bumps on stress distribution, five probe nodes were set near the maximum stress position in the M1 layer, as shown in Figure 28d. The stress components of each probe node are shown in Table 2, which provides several insights. Firstly, for nodes N1 and N5, located under the ubump, the shear and normal stresses in the Z-direction were significantly larger than those of other probe nodes. This highlights the substantial impact of the adjacent vias and bumps on the Z stress components. Secondly, node N1, positioned far from the dielectric region where the trace area was continuous, exhibited large normal stress in the X and Y directions. The combination of in-plan stress and Z stress components caused a significant increase in the local stress. Furthermore, node N3, located within the local dielectric region, had much smaller normal stress in the X and Y directions than N1, suggesting that the discontinuity of the trace area resulted in a substantial reduction in the in-plane normal stress. Finally, node N5, situated beside the long dielectric region in the X-direction, had significantly smaller Y-direction normal stress than N4, indicating the substantial influence of the directional dielectric region on the stress perpendicular to the region’s direction.

The proposed method can obtain the structure response of RDL concerning the layout impact. It is useful not only for quick identification under high-level stress but also for providing reliable loads and boundary conditions for fine mesh assessments. It enables efficient DFR within the iterative process of advanced package design.

## 4. Conclusions

This paper proposes a machine learning (ML)-based RDL modeling and simulation method for DFR of 2.5D/3D advanced packages, taking the critical layout impact into consideration. In the method, RDL was divided into blocks and subdivided into pixels of metal percentage, such that the RDL could be digitalized as metal percentage tensors. Consequently, an ANN-based surrogate model was built and trained with a small subset of tensors to predict the equivalent material properties of each block. Lastly, all blocks were transformed into elements to build an FEA model for simulations. For the validation of accuracy, line bending simulations were conducted for a 21.6 × 21.6 mm^2^ RDL using the proposed method, where the reaction force was compared with a benchmark result obtained with detailed fine mesh simulation. The results show that through the proposed method, the reaction force error can be as low as 2.81% and the layout impact can be accurately considered with 200 × 200 elements, just 1/15 of the traditional approach. The results also show that if the layout impact is neglected, the reaction force error will increase sharply as the substrate thins. Therefore, the impact must be considered in advanced package simulation. As an application case, a simulation was introduced to investigate the thermo-mechanical response of a 2.5D-integrated CPU chip at the maximum temperature state in TCT. The results indicate that for advanced packages, the maximum stress is more likely to occur in the RDL, which is different from the traditional integrations, where the maximum stress occurs in the bumps. It also reveals that the stress factors of both the RDL and bumps are significantly impacted by the layout. Moreover, it is observed that the stress in RDL is particularly impacted by vias and bumps in adjacent layers. The proposed method precisely concerns the layout-related impacts with minimal resources, presenting an opportunity to enhance the efficiency of advanced package DFR.

## Figures and Tables

**Figure 1 micromachines-14-01531-f001:**
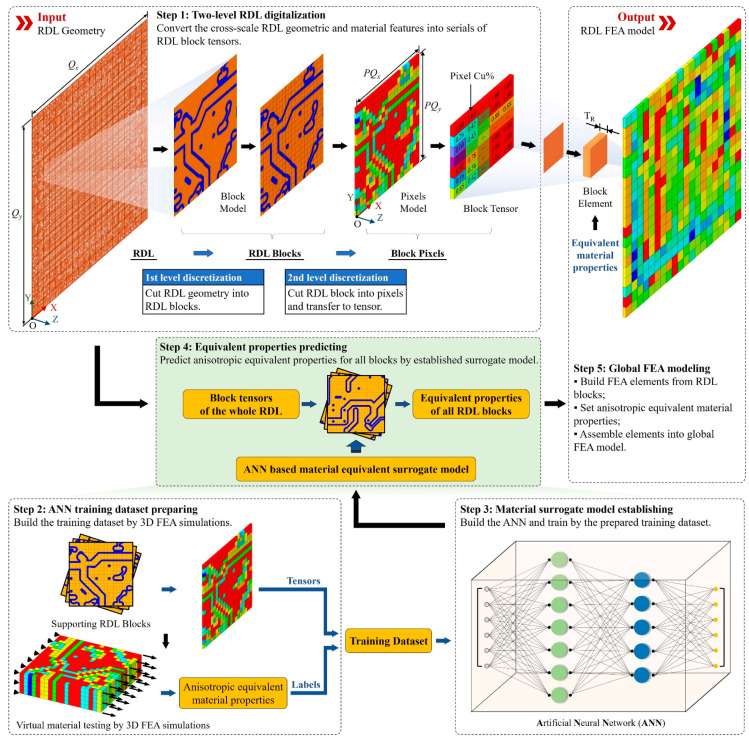
Workflow of the proposed ML-based RDL modeling method.

**Figure 2 micromachines-14-01531-f002:**
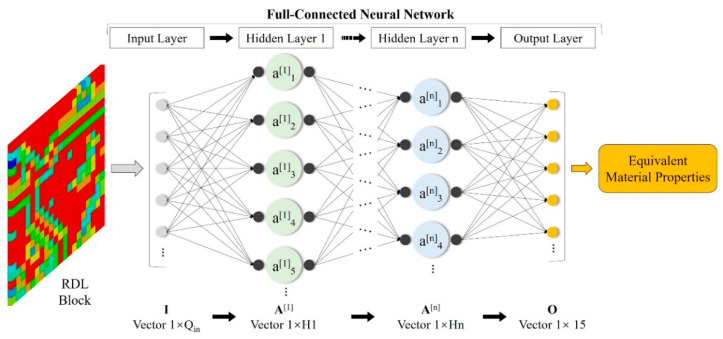
Full-connected neural network (FCNN) architecture of the proposed method (The color cubes in the RDL block are pixels with different Cu%).

**Figure 3 micromachines-14-01531-f003:**
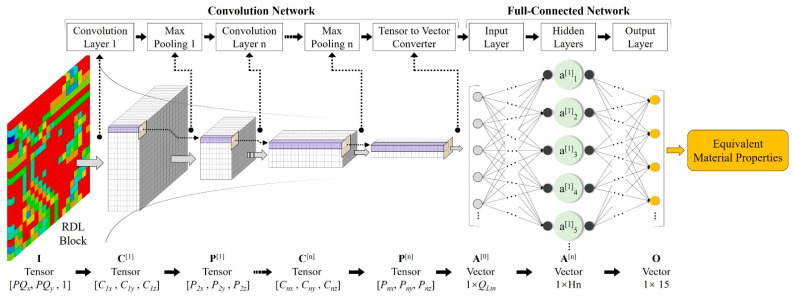
Convolution neural network (CNN) architecture of the proposed method (The color cubes in the RDL block are pixels with different Cu%).

**Figure 4 micromachines-14-01531-f004:**
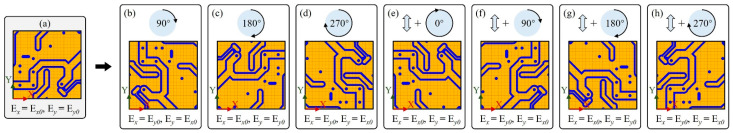
RDL block augmentation. (**a**) Initial block, (**b**) rotating of 90°, (**c**) rotating of 180°, (**d**) rotating of 270°, (**e**) initial block and flipped in Y, (**f**) rotating of 90° and flipping in Y, (**g**) rotating of 180° and flipping in Y, (**h**) rotating of 270° and flipping in Y.

**Figure 5 micromachines-14-01531-f005:**
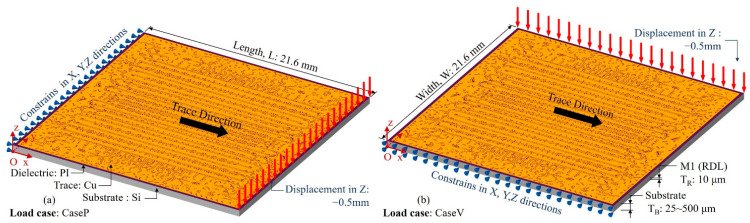
Model and load cases of the two-layered model bending simulations. (**a**) Parallel trace bending load case (Case P), (**b**) vertical trace bending load case (Case V).

**Figure 6 micromachines-14-01531-f006:**
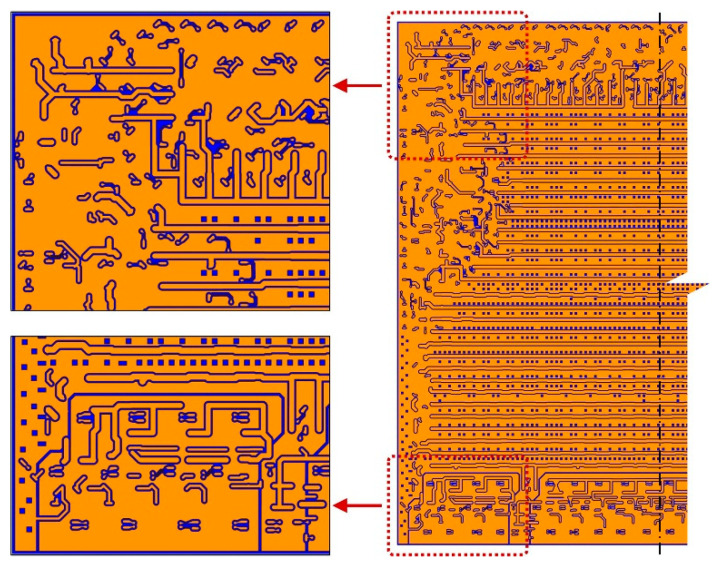
Layout pattern of M1.

**Figure 7 micromachines-14-01531-f007:**
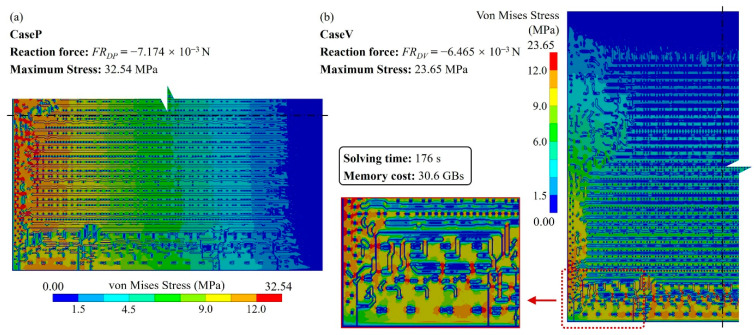
Von Mises stress distribution of the detailed fine mesh model (the substrate thickness, TB = 50 μm). (**a**) CaseP, (**b**) CaseV.

**Figure 8 micromachines-14-01531-f008:**
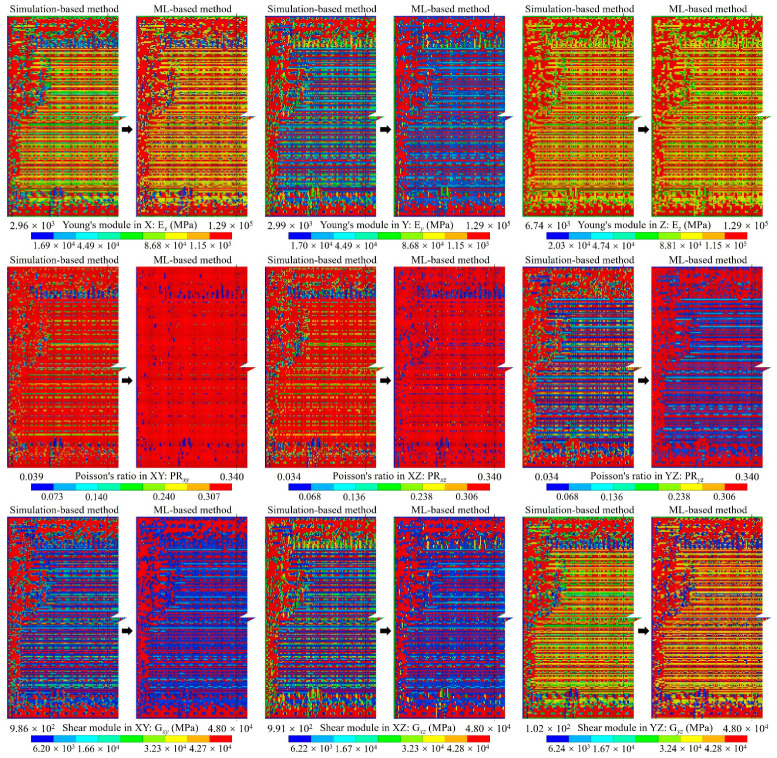
The anisotropic material of the RDL calculated using simulation-based and ML-based methods.

**Figure 9 micromachines-14-01531-f009:**
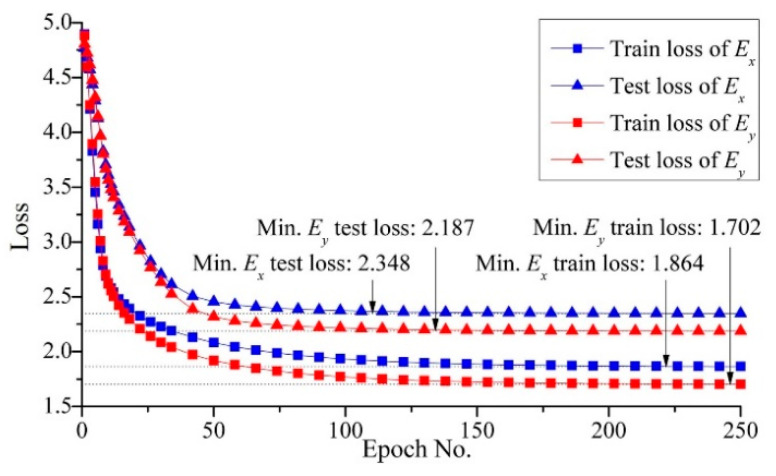
The loss variation during ANN training and the testing procedure.

**Figure 10 micromachines-14-01531-f010:**
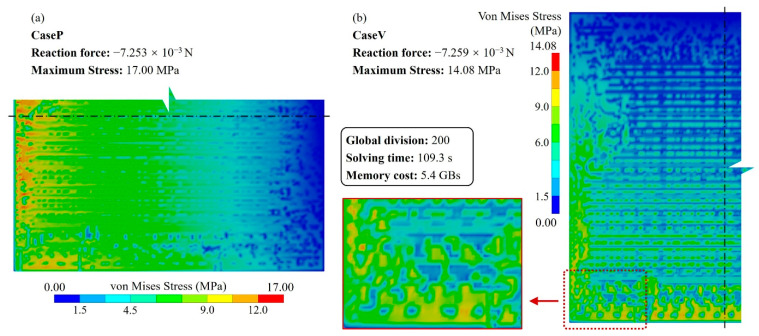
Von Mises stress distribution of the proposed ML-based modeling method (the substrate thickness, T_B_ = 50 μm). (**a**) CaseP, (**b**) CaseV.

**Figure 11 micromachines-14-01531-f011:**
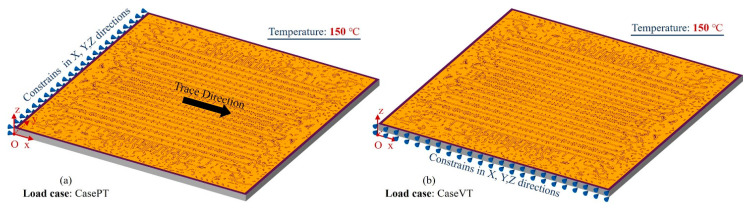
Boundary conditions and loads for two-layered model warpage simulations.

**Figure 12 micromachines-14-01531-f012:**
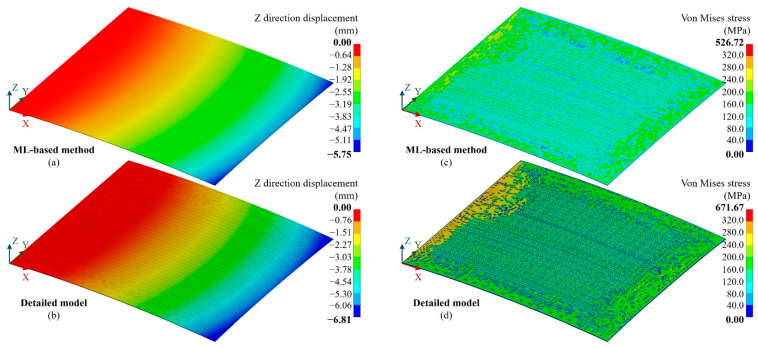
Result for the parallel trace warpage case (CasePT). (**a**) Z-direction displacement distribution of the proposed ML-based method, (**b**) Z-direction displacement distribution of the detailed fine mesh model, (**c**) von Mises stress distribution of the proposed ML-based method, (**d**) von Mises stress distribution of the detailed fine mesh model.

**Figure 13 micromachines-14-01531-f013:**
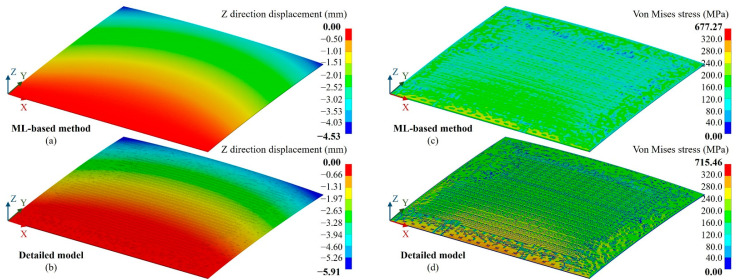
Result for the vertical trace warpage case (CaseVT). (**a**) Z-direction displacement distribution of the proposed ML-based method, (**b**) Z-direction displacement distribution of the detailed fine mesh model, (**c**) von Mises stress distribution of the proposed ML-based method, (**d**) von Mises stress distribution of the detailed fine mesh model.

**Figure 14 micromachines-14-01531-f014:**
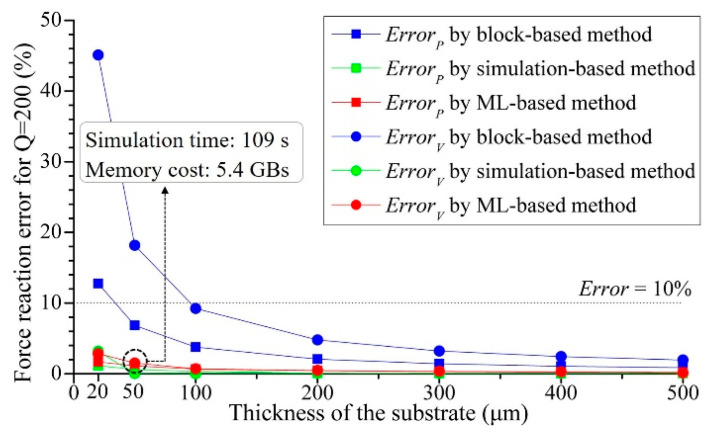
Reaction force errors of CaseP and CaseV variation by substrate thickness for different modeling methods when global mesh division *Q* = 200.

**Figure 15 micromachines-14-01531-f015:**
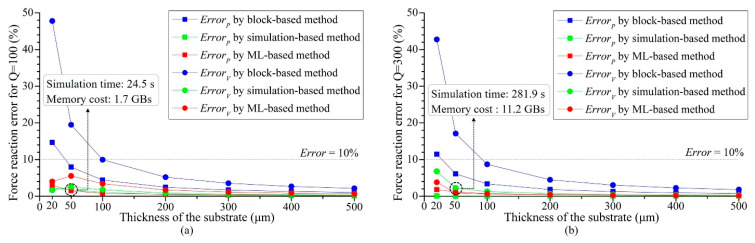
Reaction force errors of CaseP and CaseV variation by substrate thickness for different global division values. (**a**) *Q* = 100, (**b**) *Q* = 300.

**Figure 16 micromachines-14-01531-f016:**
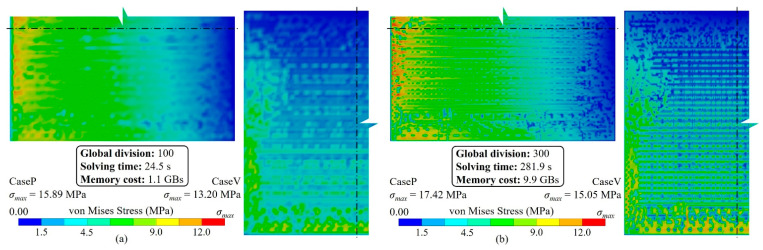
Von Mises stress distribution of different global mesh division values (the substrate thickness, T_B_ = 50 μm). (**a**) *Q* = 100, (**b**) *Q* = 300.

**Figure 17 micromachines-14-01531-f017:**
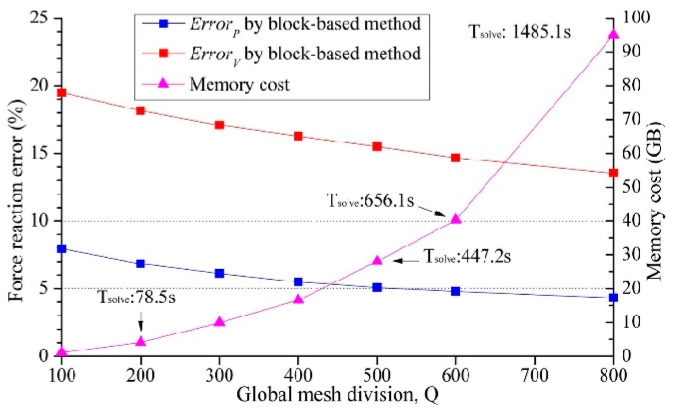
Reaction force errors and simulation costs of different global mesh division values for the traditional layout-neglecting block-based method when the thickness of the substrate equals 50 μm.

**Figure 18 micromachines-14-01531-f018:**
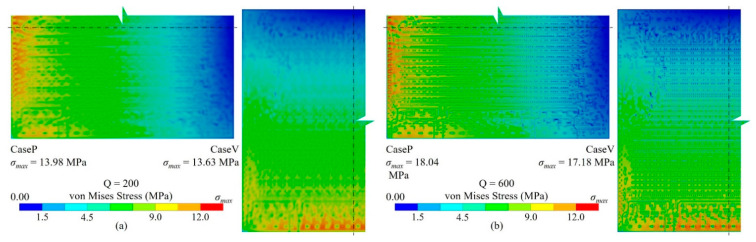
Von Mises stress distribution of the traditional layout-neglecting block-based modeling method of different global mesh division values (T_B_ = 50 μm). (**a**) *Q* = 200, (**b**) *Q* = 600.

**Figure 19 micromachines-14-01531-f019:**
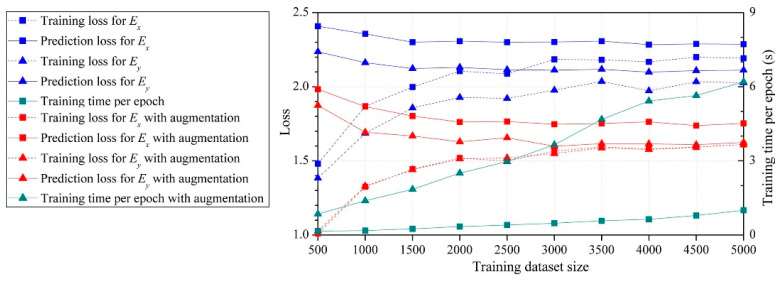
Loss and training time for different training dataset sizes.

**Figure 20 micromachines-14-01531-f020:**
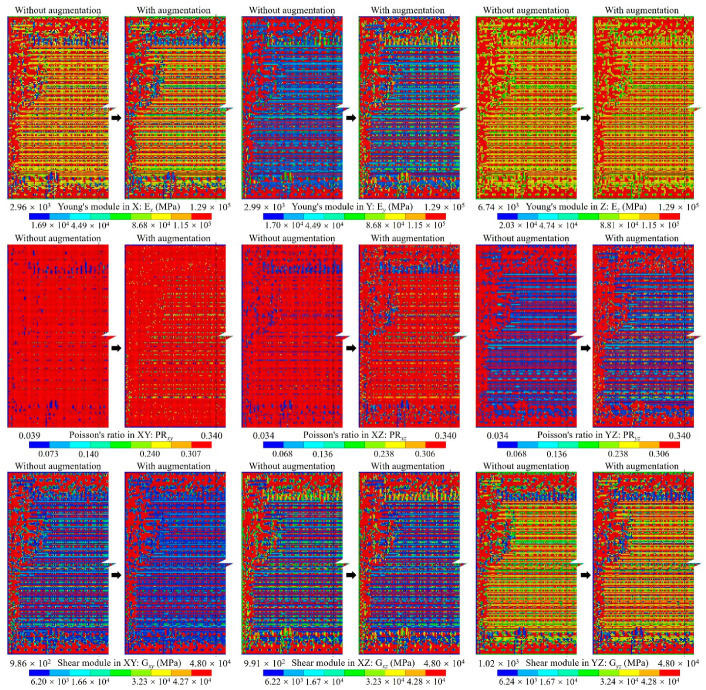
The anisotropic material of the RDL predicted by the ML-based method with training dataset augmentation.

**Figure 21 micromachines-14-01531-f021:**
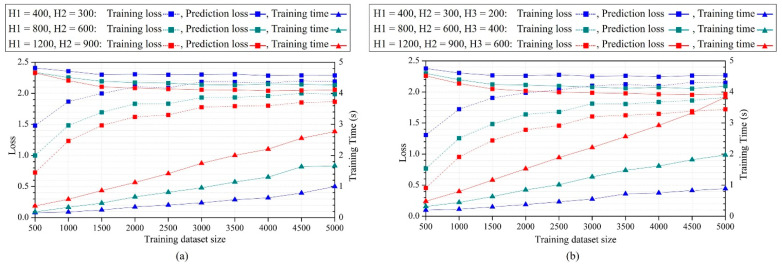
Training and prediction results of FCNNs. (**a**) FCNNs with 2 hidden layers, (**b**) FCNNs with 3 hidden layers.

**Figure 22 micromachines-14-01531-f022:**
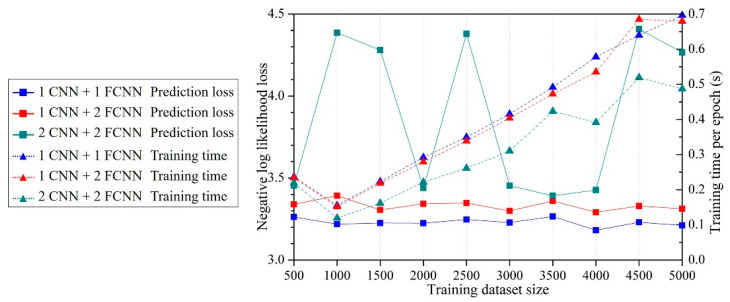
Prediction and training results of different CNNs.

**Figure 23 micromachines-14-01531-f023:**
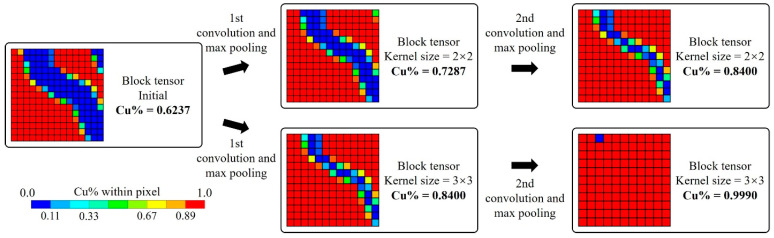
Block tensor transformation during convolution and max pooling operations.

**Figure 24 micromachines-14-01531-f024:**
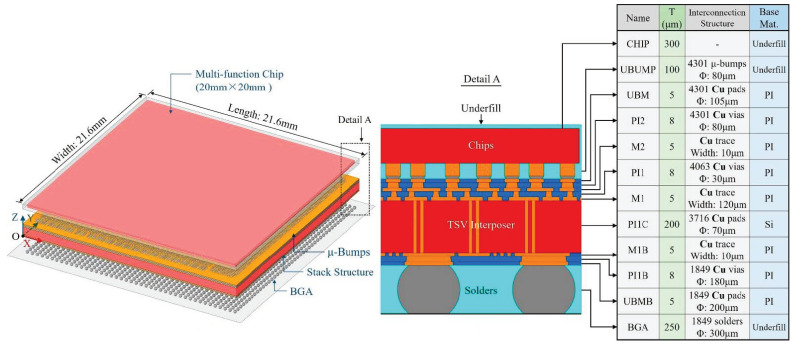
Geometry model of the 2.5D-integrated CPU chip as an application.

**Figure 25 micromachines-14-01531-f025:**
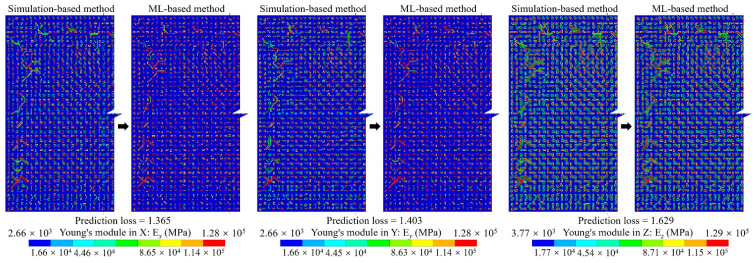
Young’s module distributions of M1B layer.

**Figure 26 micromachines-14-01531-f026:**
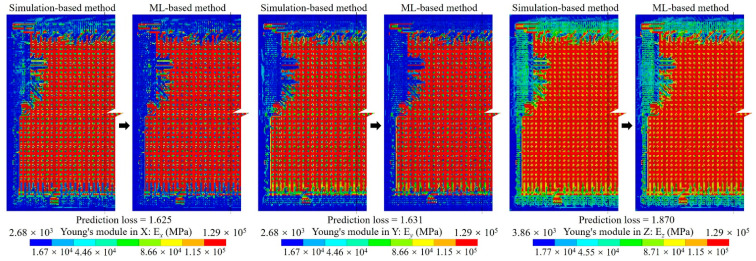
Young’s module distributions of M2 layer.

**Figure 27 micromachines-14-01531-f027:**
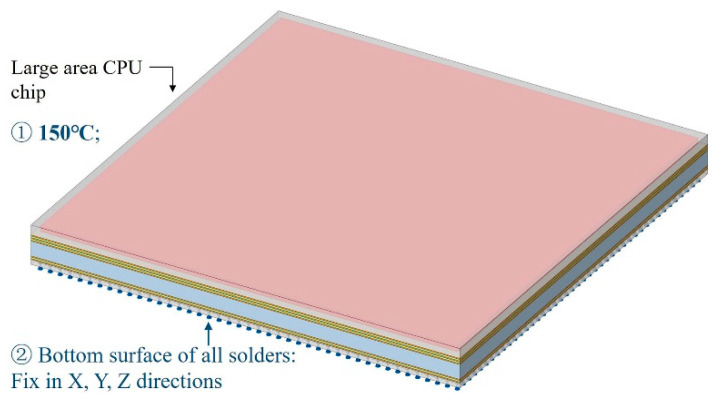
The maximum temperature state in TCT for the 2.5D-integrated CPU chip.

**Figure 28 micromachines-14-01531-f028:**
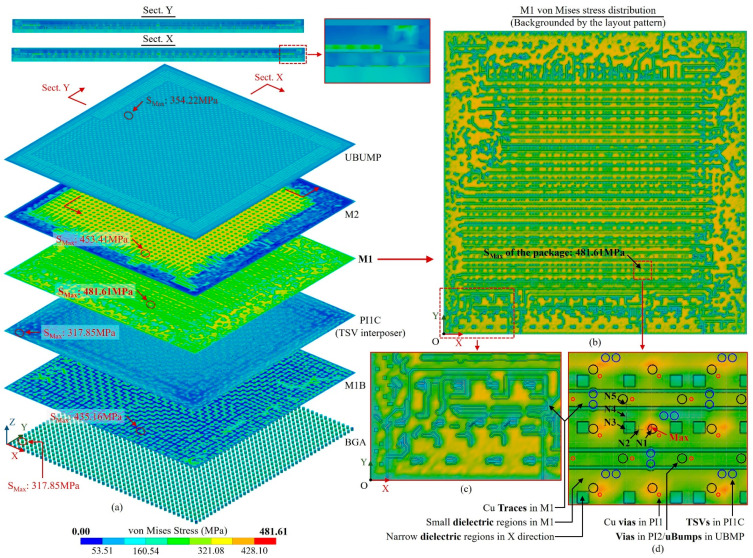
Von Mises stress distribution of the maximum temperature state of the TCT of the 2.5D-integrated CPU chip. (**a**) Key layers’ stress distributions, (**b**) M1 stress distribution backgrounded by the layout pattern, (**c**) part of the M1 stress distribution, (**d**) detail of the maximum stress region backgrounded by the M1 layout pattern and the vias and bumps in adjacent layers.

**Table 1 micromachines-14-01531-t001:** Material properties used in simulations.

Name	Si [42]	Cu [43]	PI [44]
CTE	2.6 × 10^−6^	1.64 × 10^−5^	2 × 10^−5^
Young’s Modulus (MPa)	1.31 × 10^−5^	1.30 × 10^−5^	2.5 × 10^−3^
Poisson Ratio	0.28	0.34	0.34

**Table 2 micromachines-14-01531-t002:** Stress components of probe nodes in M1 layer.

Probe Nodes	X Normal Stress (MPa)	Y Normal Stress (MPa)	Z Normal Stress (MPa)	XY Shear Stress (MPa)	YZ Shear Stress (MPa)	XZ Shear Stress (MPa)	Location
N1	−392.25	−394.70	101.78	−1.31	−0.66	−8.05	Metal under ubump
N2	−250.24	−230.09	1.77	0.10	−3.08	3.39	Metal beside small dielectric region
N3	−109.31	−103.23	32.40	0.18	−1.74	1.36	Inside local dielectric region
N4	−180.49	−117.92	−3.94	0.40	−6.37	1.17	Metal beside local dielectric region
N5	−188.31	−46.89	84.21	0.18	−5.46	0.97	Metal beside dielectric region in X direction, under ubump

## Data Availability

The data presented in this study are available on request from the corresponding author. The data are not publicly available due to privacy.

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
