# Peer review of "An RDL Modeling and Thermo-Mechanical Simulation Method of 2.5D/3D Advanced Package Considering the Layout Impact Based on Machine Learning"

_micromachines, 2023, doi:10.3390/mi14081531_

Round 1

Reviewer 1 Report

This study proposes the simulation study on thermomechanical interaction of RDL with via/bump. This is a timely topic that combine advanced packaging with machine learning. Even though it stresses that ML modeling is cost-saving than FEA, it can be remommended for publication with appropriate modifications. Some comments for revisions are given below;

1. In Figure 20, more detailed caption description such as color bar, meaning of symbol(Ex,y,z, PRxy,xz,yz) are required.

2. Several spelling and grammar should be checked in Line 92, 427~429 and so on. 

3. According to Fig. 25, Line 395~402 and conclusion, the autor stresses the stress interaction between RDL and under via/bump, but detailed reason is not suggested. Theoretical and/or experiment explanations should be provided. 

4. Section 3.3, specify the layer name of 3 RDL.(it is regarded as M1B, M1, M2). If Figure 25 is obtained by FCNN, the comparison with FEA model in terms of modeling time and accuracy would be more meaningful to prove the effectiveness of the FCNN at the overall advanced packaging structure.  

English is properly written. Please correct some spelling and grammartical errors.

Author Response

Dear reviewer:
Thanks for your letter and for reviews’ comments concerning our manuscript entitled “A RDL modeling and thermo-mechanical simulation method of 2.5D/3D advanced package considering the layout impact based on machine learning”. Those comments are all valuable and helpful for revising and improving out paper. We have studied all comments carefully and have made conscientious correction. Revised portion are marked in red in the manuscript.
The main corrections in the paper and the response to the reviews’ comments are as following:

This study proposes the simulation study on thermomechanical interaction of RDL with via/bump. This is a timely topic that combine advanced packaging with machine learning. Even though it stresses that ML modeling is cost-saving than FEA, it can be recommended for publication with appropriate modifications. Some comments for revisions are given below;
Comment 1:
In Figure 20, more detailed caption description such as color bar, meaning of symbol (Ex,y,z, PRxy,xz,yz) are required.
Response:
Thank you for your advice for the manuscript.
As your advice, we replace the Figure 9 and Figure 20 in the manuscript by the following figures.

Fig. 1 The “Figure 9. The anisotropic material of the RDL calculated by simulation-based and ML-based methods.” of the revised manuscript

Fig. 2 The “Figure 20. The anisotropic material of the RDL predicted by the ML-based method with training dataset augmentation.” of the revised manuscript

Comment 2:
Several spelling and grammar should be checked in Line 92, 427~429 and so on. 
Response: 
Thanks very much for picking up the spelling and grammar faults of the manuscript.
As your advice, we revise the following statement in the section 1.
“Regarding 2D, Liu et al. [37], Ye et al. [38] and Gong et al. [39] used high-resolution gray-scale matrix to digitalize the planar structures, then constructed convolutional neural networks (CNNs) to quickly predict the equivalent mechanical properties.”

We also revise the following statement in the section 4.
“The proposed method precisely concerns the layout-related impacts with minimal resources, presenting an opportunity to enhance the efficiency of advanced package DFR.”

Comment 3:
According to Fig. 25, Line 395~402 and conclusion, the author stresses the stress interaction between RDL and under via/bump, but detailed reason is not suggested. Theoretical and/or experiment explanations should be provided. 
Response: 
Thanks very much for your advice. 
As your question, we make following revisions in the section 3.3 to explain the reaction between RDL and under via/bump:
(1) We replace the Figure 25 in the initial manuscript by the following figure. In the revised figure we add 5 probe nodes for analyzing the stress components of RDL.

Fig. 3 The “Figure 28. Von Mises stress distribution of the maximum temperature state of the TCT of the 2.5D integrated CPU chip” in the revised manuscript
(2) We add following statement in the section 3.3 to explain the impact of the layout and vias/bumps by analyzing the stress composition of the probe nodes.
“To investigate the reasons behind the local influence of RDL geometric feature, adjacent vias and bumps on stress distribution, five probe nodes are set near the maximum stress position in the M1 layer as shown in Fig. 28(d). The stress components of each probe node are shown in Table 2, which provide several insights. Firstly, for nodes N1 and N5, located under the ubump, the shear and normal stresses in the Z-direction are significantly larger than those of other probe nodes. This highlights the substantial impact of the adjacent vias and bumps on the Z stress components. Secondly, node N1, positioned far from the dielectric region where the trace area is continuous, exhibits large normal stress in the X and Y directions. The combination of in-plan stress and Z stress components causes a significant increase of the local stress. Furthermore, node N3, located within the local dielectric region, has much smaller normal stress in the X and Y directions than N1, suggesting that the discontinuity of the trace area results in a substantial reduction in the in-plane normal stress. Finally, node N5, situated beside the long dielectric region in the X-direction, has significantly smaller Y-direction normal stress than N4, indicating the substantial influence of the directional dielectric region on the stress perpendicular to the region's direction. 
The proposed method can obtain the structure response of RDL concerning the layout impact. It is useful not only for quick identification in high-level stress but also provides reliable loads and boundary conditions for fine mesh assessments. It enables efficient DFR within the iterative process of advanced package design.”.
(3) We add the following table in the section 3.3 to illustrate the stress composition of the probe nodes.
Table 2. Stress components of probe nodes in M1 layer.
Till here, we hope this revision has answered the reviewer’s question and hope it’s acceptable. 

Comment 4:
Section 3.3, specify the layer name of 3 RDL. (It is regarded as M1B, M1, M2). If Figure 25 is obtained by FCNN, the comparison with FEA model in terms of modeling time and accuracy would be more meaningful to prove the effectiveness of the FCNN at the overall advanced packaging structure. 
Response:
We have completed the following work about this issue during our research:
Firstly, for comparing the solving efficiency of the package level model, we established a package model based on the fine mesh method. The model contains 3,588,281 nodes and 2,010,823 elements, the memory consumption for solution is more than 128GBs, which is beyond the computational capability of our hardware. The model established by the proposed method contains only 1,072,658 nodes and 625,648 elements, which cost only 3318s of time and 55GBs of memory to solve. By comparing the model scale and memory consumption, the solving efficiency is verified.
Secondly, for comparing the prediction accuracy, we calculated all equivalent material properties of all RDL blocks using the traditional simulation-based method as a benchmark. By comparing the equivalent material property distribution with the benchmark, a good consistency is shown, which verifies the prediction accuracy of the proposed method.
As your suggestion, we make the following revisions to section 3.3:
(1) We add following statement to compare the equivalent material properties calculation time and accuracy between the traditional simulation-based method and the proposed method.
“The equivalent Young's module distributions of the M1B and M2 layers are shown in Fig. 25 and Fig. 26, respectively. A good consistency with the simulation-based method is shown. The equivalent material property prediction time for the M1B and M2 layers is only 2.8211s and 2.833s respectively.”
(2) We add following figures to prove the equivalent material properties prediction accuracy of the proposed method.

Fig. 4 The “Figure 25. Young's module distributions of M1B layer.” of the revised manuscript

Fig. 5 The “Figure 26. Young's module of M2 layer.” of the revised manuscript

Reviewer 2 Report

The paper proposes a novel and potentially valuable method for improving DFR of advanced packages. With some improvements in clarity and additional detail about their methods, this could be a considerable contribution to the field. The manuscript appears to be quite comprehensive and cites relevant literature to provide background and justify the need for their work. The authors clearly describe the existing methods and identify their limitations. They present a novel approach using machine learning to address these limitations. I suggest this paper could be accepted after major revisions.

Comments:

The manuscript appears to be quite comprehensive and cites relevant literature to provide background and justify the need for their work. The authors clearly describe the existing methods and identify their limitations. They present a novel approach using machine learning to address these limitations. However, the paper need review more recent real application using deep learning. For example,  Soft Computing-based Predictive Modeling of Flexible Electrohydrodynamic Pumps, Fluidic rolling robot using voltage-driven oscillating liquid or Modeling Fabric-type Actuator Using Point Clouds by Deep Learning.

In Step 5, how were the hexahedral solid elements constructed? Were they all the same size? Did their size or shape depend on the features of the RDL block?

The authors discuss the effect of substrate thickness, global mesh division, training dataset size, and ANN architecture. However, the exact impact of these parameters on the model's predictions could be elaborated more clearly. For instance, what exactly does it mean when the authors say that "increasing dataset size generally results in improved prediction accuracy, although the effect becomes negligible beyond 2000 data points"?

The authors could also discuss why the CNNs exhibit larger losses and how this impacts the results.

The results from the large area 2.5D integrated CPU chip thermo-mechanical simulation are well presented and the authors make insightful observations from the results. However, it would be helpful to provide a more detailed explanation of these results and their implications. For instance, what does it mean for the thermo-mechanical risk of RDL to become more significant? How does this impact the design of advanced packages?

Author Response

Dear reviewer:
Thanks for your letter and for reviews’ comments concerning our manuscript entitled “A RDL modeling and thermo-mechanical simulation method of 2.5D/3D advanced package considering the layout impact based on machine learning”. Those comments are all valuable and helpful for revising and improving out paper. We have studied all comments carefully and have made conscientious correction. Revised portion are marked in red in the manuscript.
The main corrections in the paper and the response to the reviews’ comments are as following:

The paper proposes a novel and potentially valuable method for improving DFR of advanced packages. With some improvements in clarity and additional detail about their methods, this could be a considerable contribution to the field. The manuscript appears to be quite comprehensive and cites relevant literature to provide background and justify the need for their work. The authors clearly describe the existing methods and identify their limitations. They present a novel approach using machine learning to address these limitations. I suggest this paper could be accepted after major revisions.:
Comment 1:
The manuscript appears to be quite comprehensive and cites relevant literature to provide background and justify the need for their work. The authors clearly describe the existing methods and identify their limitations. They present a novel approach using machine learning to address these limitations. However, the paper need review more recent real application using deep learning. For example, Soft Computing-based Predictive Modeling of Flexible Electrohydrodynamic Pumps, Fluidic rolling robot using voltage-driven oscillating liquid or Modeling Fabric-type Actuator Using Point Clouds by Deep Learning.
Response:
Machine learning has many applications in the design and control of robots and machines. So based on your advice, we make following revisions in the section 1:
(1) We add following statement in the machine learning method introduction:
“Recently, machine learning (ML) has shown the capability of extracting interpretable models from scientific data automatically [32], it has been increasingly employed in the design and control of robots [33], actuators [34] and pumps [35]. Material equivalization of composite materials in 3D and 2D structures is a major aspect of ML research.”.
(2) We add following references:
“33.    Mao, Z.; Asai, Y.; Yamanoi, A.; Seki, Y.; Wiranata, A.; Minaminosono, A. Fluidic Rolling Robot Using Voltage-Driven Oscil-lating Liquid. Smart Mater. Struct. 2022, 31, 105006, doi:10.1088/1361-665X/ac895a.
34.    Peng, Y.; Yamaguchi, H.; Funabora, Y.; Doki, S. Modeling Fabric-Type Actuator Using Point Clouds by Deep Learning. IEEE Access 2022, 10, 94363–94375, doi:10.1109/ACCESS.2022.3204652.
35.    Mao, Z.; Peng, Y.; Hu, C.; Ding, R.; Yamada, Y.; Maeda, S. Soft Computing-Based Predictive Modeling of Flexible Electrohy-drodynamic Pumps. Biomimetic Intelligence and Robotics 2023, 100114, doi:10.1016/j.birob.2023.100114.”.
(3) We update the reference number of the whole manuscript.

Comment 2: 
In Step 5, how were the hexahedral solid elements constructed? Were they all the same size? Did their size or shape depend on the features of the RDL block?
Response: 
Thanks for your question about the element establishing method and the element shape.
In the proposed method, the RDL element creating process involves following steps: firstly, converting the rectangular face of the RDL block into a planar quadrilateral element, then sweeping the planar element in the direction of the RDL's thickness to form a solid hexahedral element, finally assigning the solid element with the equivalent material properties calculated by the material surrogate model.
Since the planar dimension of all RDL blocks are the same and the sweeping height are all equal to the thickness of the RDL, the size of each element is entirely consistent. The size and shape of the RDL elements are controlled to be not affected by the RDL features. We account for the impact of the RDL layout by assigning different equivalent material properties to each element.
Based on your suggestion, to more clearly demonstrate the modeling process, we have made the following revisions to the Section 2:
(1) We revise the Figure 1 of the manuscript, showing the procedure of creating block elements by sweeping face elements.

Fig. 6 The “Figure 1. Workflow of the proposed ML-based RDL modeling method.” of the revised manuscript
(2) We add following statement in the section 2 Step 5.
“Each element is built by sweeping the geometry face of the RDL block, with the sweep length equal to the RDL thickness, therefore all elements are in the same size.”. 
Comment 3: 
The authors discuss the effect of substrate thickness, global mesh division, training dataset size, and ANN architecture. However, the exact impact of these parameters on the model's predictions could be elaborated more clearly. For instance, what exactly does it mean when the authors say that "increasing dataset size generally results in improved prediction accuracy, although the effect becomes negligible beyond 2000 data points"?
Response: 
Thanks for your question about the “Key factors influence” part.
As your advice, we make following revisions to the section 3.2.
(1) We believe that reducing the substrate thickness will require higher modeling accuracy for RDL, but it will not affect the prediction accuracy.
(2) We revise following statement in the “The influence of the global mesh division.” part.
“Another two global division values of 100 and 300 are conducted to investigate the influence of the global division, the reaction force errors of different cases are summarized in Fig. 15. The reaction force error of CaseP is 2.255% when Q = 100, it falls to 1.149% when Q = 200, and only 0.076% when Q = 300, illustrating that larger global mesh divisions can improve solution accuracy. when Q increases from 100 to 300, the solution time increases by a factor of 11.5 and the memory occupation increases by a factor of 6.59. This suggests that a higher global mesh division can achieve a lower error, but it also increases the solution time and resource consumption.”.
(3) We revise following statement in the “The influence of the training dataset size.” part.
“ANNs was established using ten different dataset sizes ranging from 500 to 5000. Fig. 19 summarizes the loss and training time. The prediction loss curves indicate that the prediction loss decreases as the size of the training dataset increases, but the effect of dataset size is diminishing. When the dataset size exceeds 2000, curves of prediction losses are nearly parallel to the X-axis, suggesting that even if the dataset size further increases the prediction accuracy would not be improved. The prediction loss curves using the dataset augmentation is lower than the curves without dataset augmentation, indicating that the proposed dataset augmentation algorithm can improve the prediction accuracy.”
(4) We revise following statement in the “The influence of the FCNN architectures.” part.
“Three different types of FCNNs with two hidden layers and varying numbers of hidden nodes are established. The training and prediction losses are summarized in Fig. 21(a). The prediction loss curves of more hidden layer nodes are lower, indicating that the increasing hidden layer nodes results in improved prediction accuracy. However, the loss reduction in H1 from 800 to 1200 is smaller than that from 400 to 800, indicating the effect is diminishing as nodes increases. Additionally, Fig. 21(b) illustrates the loss curves of the FCNNs with three hidden layers, showing that additional hidden layer can improve accuracy but it requires more training time.”

Comment 4: 
The authors could also discuss why the CNNs exhibit larger losses and how this impacts the results.
Response: 
Thanks for your question about the CNNs part.
We believe CNNs are not suitable for the proposed method, this is due to the relatively low pixel resolution of the RDL block tensors, where convolution and pooling operations would cause undesirable defeaturing, subsequently resulting in a decrease in prediction accuracy.
As your suggestion, we have made following revisions in the section 3.2, to explain the reason that the CNNs exhibit larger losses.
(1) We add following figure in the “Application of the CNN architectures.” part.

Fig. 7 The “Figure 23. Block tensor transformation during convolution and max pooling operations.” of the revised manuscript
(2) We revise following statement in the “Application of the CNN architectures.” part.
“Fig. 22 summarizes the losses and training time for “1 CNN + 1 FCNN”, “1 CNN + 2 FCNNs” and “2 CNNs + 2 FCNNs”. The results demonstrate that while CNNs have shorter training time, they exhibit significantly larger losses across all three architectures. The intermediate tensors during convolution and max pooling operations are as shown in Fig. 23, which indicate the fact that the key features of the initial block tensor were eliminated after those operations. The undesirable defeaturing has a negative impact on the prediction accuracy.”

Comment 5: 
The results from the large area 2.5D integrated CPU chip thermo-mechanical simulation are well presented and the authors make insightful observations from the results. However, it would be helpful to provide a more detailed explanation of these results and their implications. For instance, what does it mean for the thermo-mechanical risk of RDL to become more significant? How does this impact the design of advanced packages?
Response: 
Thanks very much for your advice. 
As your question, we make following revisions in the section 3.3 to explain the reaction between RDL and under via/bump:
(1) We replace the Figure 25 in the initial manuscript by the following figure. In the revised figure we add 5 probe nodes for analyzing the stress components of RDL.

Fig. 3 The “Figure 28. Von Mises stress distribution of the maximum temperature state of the TCT of the 2.5D integrated CPU chip” in the revised manuscript
(2) We add following statement in the section 3.3 to explain the impact of the layout and vias/bumps by analyzing the stress composition of the probe nodes.
“To investigate the reasons behind the local influence of RDL geometric feature, adjacent vias and bumps on stress distribution, five probe nodes are set near the maximum stress position in the M1 layer as shown in Fig. 28(d). The stress components of each probe node are shown in Table 2, which provide several insights. Firstly, for nodes N1 and N5, located under the ubump, the shear and normal stresses in the Z-direction are significantly larger than those of other probe nodes. This highlights the substantial impact of the adjacent vias and bumps on the Z stress components. Secondly, node N1, positioned far from the dielectric region where the trace area is continuous, exhibits large normal stress in the X and Y directions. The combination of in-plan stress and Z stress components causes a significant increase of the local stress. Furthermore, node N3, located within the local dielectric region, has much smaller normal stress in the X and Y directions than N1, suggesting that the discontinuity of the trace area results in a substantial reduction in the in-plane normal stress. Finally, node N5, situated beside the long dielectric region in the X-direction, has significantly smaller Y-direction normal stress than N4, indicating the substantial influence of the directional dielectric region on the stress perpendicular to the region's direction. 
The proposed method can obtain the structure response of RDL concerning the layout impact. It is useful not only for quick identification in high-level stress but also provides reliable loads and boundary conditions for fine mesh assessments. It enables efficient DFR within the iterative process of advanced package design.”.
(3) We add the following table in the section 3.3 to illustrate the stress composition of the probe nodes.
Table 2. Stress components of probe nodes in M1 layer.

About your question on the implications of the results, we believe that the relationship between RDL stress and the reliability should be studied first. Once the relationship is established, based on the stress map obtained by our work, it can be decided whether modification of layout should be done for the high-level stress area. It is a valuable and challenging research topic, while the failure mode and reliability of RDL is mainly depended on its materials, process of fabrication, and specific geometric shapes. However, these are not the focus of this paper. The research presented in this paper focuses on creating an efficient and accurate RDL modeling method, which can quickly obtain structural responses considering the global influence of RDL to identify the high-level stress regions.

Round 2

Reviewer 1 Report

Most of obscure things are cleared based on given comments. If several spelling errors(Young's module -> Young's modulus) are corrected, it can be published with minor revised format. 

Its English is moderately written following the guideline of Micromachine.

Reviewer 2 Report

The authors diligently addressed my inquiries and incorporated abundant content into their manuscript, ensuring clarity for the readers. Thank you. I do not have any additional questions.